# MetaClass, a Comprehensive Classification System for Predicting the Occurrence of Metabolic Reactions Based on the MetaQSAR Database

**DOI:** 10.3390/molecules26195857

**Published:** 2021-09-27

**Authors:** Angelica Mazzolari, Alice Scaccabarozzi, Giulio Vistoli, Alessandro Pedretti

**Affiliations:** Dipartimento di Scienze Farmaceutiche, Università degli Studi di Milano, Via Luigi Mangiagalli 25, I-20133 Milano, Italy; angelica.mazzolari@unimi.it (A.M.); ali94-forever@hotmail.it (A.S.); giulio.vistoli@unimi.it (G.V.)

**Keywords:** drug metabolism, MetaQSAR, metabolic reactions, metabolism prediction, classification algorithms, random forest

## Abstract

(1) Background: Machine learning algorithms are finding fruitful applications in predicting the ADME profile of new molecules, with a particular focus on metabolism predictions. However, the development of comprehensive metabolism predictors is hampered by the lack of highly accurate metabolic resources. Hence, we recently proposed a manually curated metabolic database (MetaQSAR), the level of accuracy of which is well suited to the development of predictive models. (2) Methods: MetaQSAR was used to extract datasets to predict the metabolic reactions subdivided into major classes, classes and subclasses. The collected datasets comprised a total of 3788 first-generation metabolic reactions. Predictive models were developed by using standard random forest algorithms and sets of physicochemical, stereo-electronic and constitutional descriptors. (3) Results: The developed models showed satisfactory performance, especially for hydrolyses and conjugations, while redox reactions were predicted with greater difficulty, which was reasonable as they depend on many complex features that are not properly encoded by the included descriptors. (4) Conclusions: The generated models allowed a precise comparison of the propensity of each metabolic reaction to be predicted and the factors affecting their predictability were discussed in detail. Overall, the study led to the development of a freely downloadable global predictor, MetaClass, which correctly predicts 80% of the reported reactions, as assessed by an explorative validation analysis on an external dataset, with an overall MCC = 0.44.

## 1. Introduction

From their advent in pharmaceutical research, machine learning (ML) algorithms have found many successful applications, ranging from hit identification to drug delivery optimization [1,2,3]. This success can be explained by considering their capacity to unveil patterns and relationships even when analyzing huge amounts of complex data [4]. Among the fields where these approaches can find relevant applications, the in silico prediction of drug metabolism is attracting great interest [5,6]. Its relevance is easily understandable by considering the ever-increasing role played by the capacity to predict the ADME/Tox profile of a given molecule starting from the early phases of drug discovery [7,8].

In general, artificial intelligence (AI) methods can find at least three major applications focused on drug metabolism [9] since they can predict: (1) the potential sites of metabolism (regardless of the involved reactions) [10], (2) the metabolic reaction(s) that a given substrate undergoes [11] and (3) the formed metabolites [12]. These three applications can be also seen as the progressive steps of an ideal workflow that allows the comprehensive prediction of the entire metabolic fate of a given compound [13]. Furthermore, AI approaches can be used to predict the toxicity profile of the parent compound and of the predicted metabolites [14].

Notwithstanding the above, most of the reported predictive studies are still focused on redox metabolism, with particular attention paid to CYP-450 catalyzed reactions [15]. In contrast, very few studies describe predictive models for the conjugation reactions and even fewer studies try to offer comprehensive predictions of the metabolic profile of a given compound [16,17]. The available global approaches are mostly knowledge-based methods, which predict the possible metabolic profile based on the occurrence of specific reactive sites or structural alerts [18].

The discrepancy between the studies focused on redox metabolic reactions and those dealing with the other metabolic reactions might be justified by considering the marked relevance of the former, which represent by far the most common biotransformations [19]. Nevertheless, this explanation is questionable and the lack of predictive models for the conjugations cannot be easily justified since they are very frequent reactions that represent crucial mechanisms of detoxification [20].

A more pertinent cause that might explain the scarcity of comprehensive predictors is the lack of extended and truly accurate metabolic databases. Most available datasets are focused on the metabolic reactions catalyzed by CYP-450 isozymes [21] and often combine xenobiotic and endogenous reactions with a metabolomics perspective [22,23,24]. Thus, they are not properly tailored to the development of general predictive systems. In addition, they are almost always collected by the automatic querying of other available resources and thus their level of accuracy is not very high [25]. This problem can have detrimental effects when performing predictive analyses since even a few inaccurate data can undermine the performance of the resulting models.

On these grounds, we recently reported a novel metabolic database (MetaQSAR), which was generated by manual and critical meta-analyses of the specialized literature published during the 2005–2015 period [26]. The collected data proved successful in developing satisfactory models for predicting the occurrence of conjugation reactions with glucuronic acid [11] and glutathione [27], which represent frequent metabolic reactions and important detoxification processes.

Based on these encouraging results, the MetaQSAR database is here exploited to perform an exhaustive predictive analysis of all the sets of metabolic reactions for which the database includes sufficient instances (i.e., ≥50). In these predictive studies, we follow the same classification system adopted by MetaQSAR, which subdivides the metabolic reactions into three main classes (i.e., redox reactions, hydrolyses and conjugations), 21 classes and 101 subclasses. In detail, the reported analyses involve the 3 main classes, 18 classes and 23 populated subclasses. Details concerning the subdivision into main classes, classes and subclasses can be found elsewhere [26]. The models were developed using the random forest classification algorithm and using sets of physicochemical, stereo-electronic and constitutional descriptors.

Besides developing comprehensive models able to predict the occurrence of the major metabolic reactions, the study aims to offer a comparative analysis of the propensity of each metabolic reaction to be predicted. Such a propensity might depend on the chemical space covered by the extracted datasets and on the informative richness of the computed descriptors, but also on the intrinsic features that govern the predicted reactions. These comparisons should reveal which reactions are unpredictable (at least by using the available data), which can be predicted even by applying standard methods, as reported here, and which reactions require targeted approaches to enhance the resulting performance. The predictive models generated for the classes of reactions are also utilized to develop a global tool, called MetaClass, and implemented within the VEGA environment [28], which predicts the occurrence of each class of metabolic reactions for a given input molecule.

## 2. Results

### 2.1. MetaQSAR-Based Datasets

The reported comparative analysis is based on the metabolic reactions collected within the MetaQSAR database and focuses on the first-generation reactions. Overall, the study involved 3788 metabolic reactions, which could be subdivided into 3 major classes, 21 classes and 101 subclasses. Predictive models were generated for the three major classes (Table 1 and Table 2) and for all the classes and the subclasses with at least 50 instances, as compiled in Table 3 and Table 4, which also include a brief description of the metabolic reactions belonging to the considered classes or subclasses. In all performed studies, the datasets were generated by considering as substrates (class S) all the molecules for which the corresponding metabolic reaction is reported and as non-substrates (class NS) all the remaining molecules without exceptions. This kind of classification poses the problem of false negatives since the lack of experimental evidence about a given metabolic reaction does not necessarily imply that that the resulting metabolite cannot occur. Indeed, the metabolic reaction might be unreported because the corresponding metabolites were undetectable with the adopted analytical methods or simply because they were not searched as the analyzed paper was designed with different objectives. In a recent study, we proposed to focus the analysis on the papers reporting exhaustive metabolic trees in order to reduce the number of false positives [27]. Nevertheless, this study considered all the metabolic reactions included in the MetaQSAR database in order to maximize the number of metabolic classes and subclasses with sufficient instances to develop reliable predictive models.

As described in the Methods, the initial predictive analyses of the three major classes were utilized in the validation phase to tune the developed models by selecting the best-performing algorithm and by optimizing the corresponding hyperparameters. Moreover, they also had a calibrating role for the following class- and subclass-specific studies since these initial analyses were carried out for evaluating (1) the performance achieved by various sets of descriptors, (2) the relevance of using balanced datasets and (3) the role of the substrate’s ionization state. The following predictive studies were then carried out by applying the derived best-performing conditions.

### 2.2. Predictive Models for the Three Main Classes

Table 1 compiles the performances of the predictive models for the three major classes of metabolic reactions, as parameterized by some well-known metrics, including both overall and class-specific parameters. In detail, these predictive studies involved both the unbalanced datasets as directly extracted from MetaQSAR and the balanced datasets as generated by random undersampling (US) of the majority class. The majority class corresponds here to the class of substrates for the redox reactions and to that of non-substrates for hydrolyses and conjugations. The performed analyses involved three sets of descriptors (as described in the Methods), which were calculated by considering the molecules in their neutral state.

**Table 1 molecules-26-05857-t001:** Performance of the predictive models for the three major classes of metabolic reactions by considering both the unbalanced and the randomly undersampled (US) balanced datasets. Three sets of descriptors were used for each analysis (S and NS stand for substrates and non-substrates, respectively; PC + Elec indicates the set of physicochemical plus stereo-electronic descriptors).

**Metrics**	**Unbalanced Datasets**						**US Balanced Datasets**					
	**Redox Metabolic Reactions (Classes 01–08)**											
	**BlueDesc**		**PC + Elec**		**Kier–Hall**		**BlueDesc**		**PC + Elec**		**Kier–Hall**	
	**NS**	**S**	**NS**	**S**	**NS**	**S**	**NS**	**S**	**NS**	**S**	**NS**	**S**
Precision	0.62	0.73	0.63	0.72	0.66	0.74	0.73	0.72	0.72	0.71	0.73	0.71
Recall	0.56	0.78	0.55	0.79	0.57	0.81	0.71	0.74	0.69	0.74	0.69	0.74
MCC	0.35		0.35		0.39		0.44		0.43		0.44	
AUC	0.73		0.72		0.76		0.78		0.77		0.80	
	**Hydrolysis Metabolic Reactions (Classes 11–14)**											
	**BlueDesc**		**PC + Elec**		**Kier–Hall**		**BlueDesc**		**PC + Elec**		**Kier–Hall**	
	**NS**	**S**	**NS**	**S**	**NS**	**S**	**NS**	**S**	**NS**	**S**	**NS**	**S**
Precision	0.91	0.69	0.90	0.67	0.91	0.74	0.74	0.75	0.74	0.74	0.75	0.78
Recall	0.97	0.42	0.97	0.34	0.98	0.44	0.76	0.74	0.74	0.74	0.79	0.74
MCC	0.48		0.42		0.52		0.49		0.48		0.53	
AUC	0.81		0.79		0.84		0.81		0.79		0.83	
	**Conjugation Metabolic Reactions (Classes 21–28)**											
	**BlueDesc**		**PC + Elec**		**Kier–Hall**		**BlueDesc**		**PC + Elec**		**Kier–Hall**	
	**NS**	**S**	**NS**	**S**	**NS**	**S**	**NS**	**S**	**NS**	**S**	**NS**	**S**
Precision	0.81	0.53	0.81	0.55	0.82	0.58	0.69	0.69	0.67	0.67	0.69	0.69
Recall	0.88	0.41	0.89	0.40	0.89	0.45	0.69	0.69	0.67	0.67	0.68	0.70
MCC	0.31		0.32		0.37		0.37		0.34		0.38	
AUC	0.72		0.73		0.77		0.73		0.71		0.75	

The comparison of the results reached by the three major classes reveals that hydrolyses afford, on average, the best performance regardless of the utilized descriptors and datasets, while conjugations yield the worst results. These findings can be explained by considering that the hydrolytic reactions, while involving different enzymes, comprise a rather homogeneous set of metabolic reactions. Hence, a single predictive model can account for all the collected reactions. In contrast, the redox reactions and especially the conjugations comprise a wide variety of metabolic reactions, which involve different enzymes, different catalytic mechanisms and different substrate preferences. Thus, their occurrence cannot be properly predicted by unique models.

Concerning the set of descriptors, Table 1 shows that the models based on the BlueDesc set, while involving a very high number of variables, afford, on average, performance comparable to that reached when considering the set of physicochemical and stereo-electronic descriptors. This finding is in line with previous studies that emphasized the key role of stereo-electronic descriptors in predicting conjugation with glutathione [27]. In all the analyses, the Kier–Hall indices provide the best predictive models. These results underline the beneficial role played by descriptors encoding for the occurrence of specific moieties (or atom types), presumably as they account for the presence of the functional groups that undergo the predicted metabolic reaction.

While the overall metrics show limited differences, the comparison of the class-specific performance reached by balanced and unbalanced datasets emphasizes the beneficial effects exerted by balancing the datasets. The beneficial effect of the balanced datasets can be appreciated by considering that the capacity to recognize the instances of the minority classes (i.e., the non-substrates for redox and the substrates for the other groups) is around 0.5 in all the analyses involving the unbalanced datasets. This means that the thus obtained predictions are comparable to the random results. In contrast, the performance of the minority class increases by using the balanced datasets, reaching rather satisfactory values at least for redox reactions and hydrolyses.

With regard to the role of the ionization state, Table 2 reports the performance as obtained by considering the molecules in their most probable ionization forms at physiological pH. Based on the above-described results, the analysis involves the balanced datasets and is focused on physicochemical and stereo-electronic descriptors as well as on the Kier–Hall topological indices. Table 2 reveals that ionization alters the performance trends compared to those obtained by using the neutral forms. In detail, the comparison of the performance reached by neutral and ionized substrates reveals that the ionization has a negative effect on the performance of the redox reactions, a negligible role for the hydrolyses, while showing a beneficial impact on conjugations.

**Table 2 molecules-26-05857-t002:** Performance of the developed predictive models for the three major classes of metabolic reactions when considering ionized substrates. Based on the previous results, the predictive analyses focused on the balanced datasets and two sets of descriptors were considered (physicochemical plus stereo-electronic descriptors and Kier–Hall topological indices). In parentheses, the differences with the metrics reported in Table 1.

Metrics	US Balanced Datasets/Ionized Substrates											
	Redox Metabolic Reactions (Classes 01–08)				Hydrolysis Metabolic Reactions (Classes 11–14)				Conjugation Metabolic Reactions (Classes 21–28)			
	Kier–Hall		PC + Elec		Kier–Hall		PC + Elec		PC + Elec		Kier–Hall	
	NS	S	NS	S	NS	S	NS	S	NS	S	NS	S
Precision	0.67	0.67	0.69	0.68	0.75	0.78	0.74	0.77	0.68	0.69	0.70	0.71
Recall	0.67	0.68	0.67	0.70	0.80	0.73	0.78	0.73	0.68	0.70	0.71	0.68
MCC	0.35 (−0.07)		0.37 (−0.07)		0.52 (−0.01)		0.47 (−0.02)		0.36 (+0.02)		0.41 (+0.03)	
AUC	0.72 (−0.05)		0.75 (−0.05)		0.81 (−0.02)		0.78 (−0.01)		0.72 (+0.01)		0.77 (+0.02)	

The Kier–Hall indices afford also here the best models, and the performance of both sets of descriptors is similarly affected by the ionization. The effect on Kier–Hall topological indices suggests that they properly capture the structural differences between neutral and ionized molecules. The effects on stereo-electronic parameters are expected and predictable, while the impact on physicochemical descriptors can be mostly ascribed to lipophilicity-related features. Since the ionization plays a negative (albeit limited) role in most comparisons (4 out of 6) and considering the possible inaccuracies introduced by automatic generation of the most plausible ionized forms, the following class-specific models were developed by considering the substrates in their neutral form.

### 2.3. Predictive Models for the Metabolic Classes

Table 3 reports the performance of the predictive models generated for the metabolic classes with at least 50 instances (18 classes out of 21). The models were generated by utilizing only the balanced datasets and the two sets of descriptors that afforded the best performance in the previous analyses, namely physicochemical plus stereo-electronic descriptors and the Kier–Hall topological indices. For the sake of simplicity, Table 3 includes only overall metrics, namely MCC and AUC values. For each major class, Table 3 also reports the average performance.

**Table 3 molecules-26-05857-t003:** Performance of the predictive models for the 21 classes of metabolic reactions based on US balanced datasets and exploiting two sets of descriptors (physicochemical plus stereo-electronic descriptors and Kier–Hall topological indices). Classes 09, 13 and 28 do not include enough instances (<50) to develop predictive models. Only overall performance metrics are compiled.

Class ID	Description	No. of Instances	PC + Elec		Kier–Hall	
			MCC	AUC	MCC	AUC
01	Oxidation of Csp^3^	1006	0.24	0.64	0.31	0.70
02	Oxidation of Csp^2^ and Csp	589	0.24	0.65	0.28	0.70
03	CHOH ↔ C=O → COOH	143	0.36	0.74	0.40	0.80
04	Various redox reactions of carbon atoms	43	---	---	---	---
05	Redox reactions of R_3_N	117	0.30	0.69	0.44	0.77
06	Redox reactions of >NH, >NOH, and –N=O	159	0.47	0.80	0.59	0.86
07	Redox of quinones or analogues	112	0.44	0.77	0.48	0.78
08	Redox of S atoms	126	0.68	0.90	0.76	0.93
09	Redox of other atoms	6	---	---	---	---
**Main class 1**	**Average**		**0.39**	**0.74**	**0.46**	**0.79**
11	Hydrolysis of esters, lactones and inorganic esters	314	0.58	0.86	0.77	0.94
12	Hydrolysis of amides, lactams and peptides	122	0.42	0.80	0.52	0.83
13	Epoxide hydration	7	---	---	---	---
14	Other hydrolyses	90	0.26	0.69	0.29	0.77
**Main class 2**	**Average**		**0.42**	**0.78**	**0.53**	**0.85**
21	*O*-glucuronidations and glycosylations	343	0.42	0.76	0.55	0.84
22	*N*- and *S*-glucuronidations/all other glycosilations	131	0.27	0.71	0.42	0.77
23	Sulfonations	114	0.35	0.75	0.52	0.83
24	GSH and RSH conjugations	171	0.54	0.85	0.68	0.89
25	Acetylations and acylations	74	0.60	0.85	0.60	0.87
26	CoASH-ligation followed by amino acid conjugations	50	0.80	0.95	0.81	0.95
27	Methylations	50	0.60	0.84	0.58	0.80
28	Other conjugations	37	---	---	---	---
**Main class 3**	**Average**		**0.51**	**0.82**	**0.59**	**0.85**

Overall, Table 3 reveals that there is no relation between performance and the number of instances in each class. This result can be justified when considering that the models were developed by using balanced datasets and suggests that the differences in the observed performance mostly depend on the intrinsic features of the predicted metabolic reactions rather than on the extent of the chemical space covered by the substrates. At most, Table 3 suggests that very populated classes (such as 01 and 02) show poor performance, which is reasonable as they include heterogeneous metabolic reactions (and substrates) that involve different reactive centers.

The analysis of the reported average performance reveals a trend in substantial disagreement with that evidenced by Table 1. Indeed, the conjugations provide the best predictive models, followed by hydrolyses, while redox reactions yield the worst average performance. These findings emphasize that the seven analyzed classes of conjugations comprise rather homogeneous metabolic reactions, thus allowing the development of reliable predictive models. In contrast, the eight monitored classes of redox reactions include more heterogeneous metabolic reactions and this negatively impacts the performance of the resulting predictive models.

The analysis of the performance reached by each specific class allows for some considerations. Concerning the redox reactions, Table 3 suggests that the reactions involving carbon atoms (MCC average = 0.37 and AUC average = 0.74, using the Kier–Hall indices) are more challenging than those affecting heteroatoms (MCC average = 0.60 and AUC average = 0.85). This difference may be explained by considering that the redox reactions on carbon atoms are more heterogeneous than those on heteroatoms, which involve a limited number of reactive groups, which can be conveniently recognized by topological descriptors. The best-performing class of redox reactions on carbon atoms is that involving quinones and analogues, which indeed comprises homogeneous (and easily detectable) reactive centers. Among the classes involving heteroatoms, the redox reactions on the sulfur atoms yield the best results, probably as they affect a well-defined set of reactive moieties.

Concerning the hydrolyses, Table 3 suggests that the average values are worsened by class 14, which includes a limited but very heterogeneous group of hydrolytic reactions. In contrast, the two most populated classes (11 and 12) show, on average, satisfactory performance, in line with that yielded by conjugations. This finding can be explained by considering that these reactions involve a limited set of labile groups. Notably, the hydrolysis of esters can be more easily predicted than that of amides.

When focusing on conjugations, Table 3 shows that the worst performance is obtained for class 22, which includes all glucuronidations not involving oxygen atoms and thus comprises various metabolic reactions. In contrast, the other classes involve well-characterized sets of reactive centers and can be conveniently predicted (MCC > 0.5, using the Kier–Hall indices). Finally, the unsatisfactory performance yielded by sulfonations (class 23) can be justified by considering the high polarity of the resulting metabolites. These polar metabolites are not easily detectable by applying standard analytical techniques and thus the mentioned problem of false negatives might markedly affect the predictive performance for this class of metabolic reactions.

With regard to the sets of descriptors, Table 3 shows that Kier–Hall indices perform better than physicochemical and stereo-electronic descriptors for 15 out of 18 classes. Only for methylations (class 27) do physicochemical and stereo-electronic descriptors perform better, and in two classes (25 and 26), the two sets show comparable performance. On average, MCC values show larger differences than AUC parameters and the MCC values reveal differences ≥0.1 in 8 out of 15 classes. Taken together, the better performance afforded by Kier–Hall indices emphasizes the relevance of properly recognizing the involved reactive groups to predict the occurrence of a given metabolic reaction.

Along with the discussed heterogeneity of the involved metabolic reactions, a factor that can explain the differences in performance reached by the analyzed classes is the efficiency of the considered metabolic reaction. Stated differently, a metabolic class can be conveniently predicted if almost all the substrates that contain a given reactive group undergo the corresponding metabolic reaction. By contrast, when the presence of a given reactive moiety is a necessary but not sufficient condition and many substrates that include such a reactive center do not undergo the metabolic reaction, its occurrence can be predicted with greater difficulty. Regardless of the heterogeneity of the involved reactive moieties, Table 3 indicates that, on average, conjugations are more easily predictable than redox reactions. This suggests that the enzymes involved in the redox reactions are less efficient, which is reasonable as their catalytic activity depends on various stereo-electronic features that go beyond the mere presence of a detectable reactive center.

### 2.4. Predictive Analyses for the Metabolic Subclasses

Table 4 compiles the predictive performance values for the subclasses of metabolic reactions with at least 50 substrates (23 out of 101) plus the corresponding average metrics. An overall analysis of the reported MCC and AUC values reveals performance that is in line with that discussed for the classes of reactions. The comparison of the average metrics between classes and subclasses reveals that a more detailed classification (as seen in subclasses) has a limited role for redox reactions, a beneficial impact on hydrolyses and a moderate effect on conjugations. These findings suggest that the difficulty in predicting redox reactions is not only ascribable to the heterogeneity of the involved biotransformations (although some subclasses are still highly populated) but rather to the complexity of the factors governing them. Indeed, the classes of conjugations already include homogeneous metabolic reactions and substrates and thus their further subdivision has a limited effect, while the hydrolyses benefit from the more precise classification of the reactions involving the ester groups.

**Table 4 molecules-26-05857-t004:** Performance of the predictive models for the subclasses of metabolic reactions with at least 50 substrates based on US balanced datasets and exploiting two sets of descriptors (physicochemical plus stereo-electronic descriptors and Kier–Hall topological indices). Only overall performance metrics are compiled.

Subclass ID	Description	No. of Instances	PC + Elec		Kier–Hall	
			MCC	AUC	MCC	AUC
01.01	Oxidations of isolated Csp^3^	193	0.32	0.72	0.43	0.83
01.02	Oxidations of C in α to an unsaturated system	233	0.20	0.64	0.29	0.68
01.03	Oxidations of Csp^3^ carrying an heteroatom	492	0.16	0.56	0.27	0.66
01.04	Dehydrogenations	81	0.26	0.67	0.32	0.70
02.01	Oxidations of aryl compounds	440	0.22	0.62	0.23	0.66
02.02	Oxidations of azarenes	94	0.39	0.72	0.38	0.78
02.03	Oxidations of >C=C<	55	0.31	0.77	0.74	0.94
03.02	Hydrogenations of carbonyls	71	0.41	0.79	0.70	0.91
05.01	Oxidations of tertiary alkylamines	65	0.19	0.67	0.60	0.81
06.01	Hydroxylations of amines	62	0.55	0.84	0.68	0.91
07.04	Oxidations of phenols	51	0.51	0.79	0.43	0.78
08.03	Oxygenations of sulfides	83	0.58	0.85	0.72	0.93
**Main class 1**	**Average**		**0.34**	**0.72**	**0.48**	**0.80**
11.01	Hydrolysis of alkyl esters	103	0.58	0.88	0.81	0.96
11.03	Hydrolysis of anionic and cationic esters	100	0.78	0.95	0.85	0.97
11.08	Hydrolysis of esters of inorganic acids	51	0.65	0.90	0.92	0.99
12.02	Hydrolysis of anilides and hydrazides	55	0.58	0.85	0.62	0.88
**Main class 2**	**Average**		**0.65**	**0.90**	**0.80**	**0.95**
21.01	*O*-glucuronidation of alcohols	85	0.59	0.84	0.70	0.91
21.02	*O*-glucuronidation of phenols	152	0.55	0.86	0.75	0.94
21.03	*O*-glucuronidation of carboxylic acids	99	0.53	0.81	0.72	0.91
22.01	*N*-glucuronidation of linear and cyclic amines	97	0.45	0.77	0.51	0.83
23.01	*O*-sulfonation of phenols	70	0.53	0.85	0.64	0.91
24.01	Nucleophilic additions of glutathione	89	0.59	0.85	0.69	0.92
24.02	Reactions of glutathione addition−elimination	68	0.50	0.81	0.63	0.88
**Main class 3**	**Average**		**0.53**	**0.83**	**0.66**	**0.90**

In detail, the achieved performance for redox reactions confirms the difficulty in predicting the biotransformations on the carbon atoms. This is partly ascribable to the richness of substrates within these subclasses, but is especially due to the variety of factors that govern these redox reactions, which cannot be described by the simple occurrence of reactive fragments. Only the subclasses of reactions involving double bonds and carbonyls can be conveniently predicted (MCC > 0.6). In addition, Table 4 confirms that redox reactions involving heteroatoms are more easily predicted.

The subclasses for the hydrolyses reach remarkable performance, especially for esters, the subdivision of which into neutral and ionizable substrates enhances the resulting performance. Table 4 confirms the greater difficulty in predicting the hydrolysis of amides and derivatives compared to that of esters. Concerning the conjugations, Table 4 evidences the beneficial role of a more precise clustering of the *O*-glucoronidations on the resulting performance, while the further classification of the reactions with GSH has a negligible role on the developed predictive models.

The overall effect of the clustering into subclasses can be appreciated by analyzing the charts reported in Figure 1, which highlight the increase in the number of models with AUC > 0.90. In detail, there is one satisfactory model per major class when analyzing the metabolic classes, while the number of highly performing models increases to four, three and five when focusing on the subclasses of redox reactions, hydrolyses and conjugations, respectively. By considering the relative abundances of the predicted classes and subclasses, Figure 1 reveals that the vast majority of redox reactions are unsatisfactorily predicted by analyzing both classes and subclasses. In contrast, the majority of hydrolyses are conveniently predicted by considering both classes and subclasses. Finally, Figure 1 evidences the beneficial effect exerted by clustering the conjugation reactions into subclasses, with enhancements particularly noticeable for glucuronidations.

With regard to the utilized sets of descriptors, Table 4 reveals results in agreement with Table 2 since the Kier–Hall indices perform better in 22 cases out of 23, and physicochemical and stereo-electronic descriptors provide better performance only in predicting the oxidation of phenols (subclass 07.04). On average, the differences in the performance yielded by the two sets of descriptors are even greater when considering the subclasses, since the MCC differences are greater than 0.1 in 13 cases out of 23. The enhanced performance provided by the Kier–Hall indices can be explained by considering that their capacity to conveniently describe the involved reactive groups increases with the specificity of the predicted reactions and the subclasses comprise more homogeneous reactions compared to the classes.

As discussed above, the performance of the Kier–Hall indices depends on the efficacy of the considered reactions. The performance achieved by the analyzed subclasses confirms the greater efficacy of hydrolyses and conjugations compared to redox reactions. Stated differently, the performance values compiled in Table 4 suggest that almost all inorganic esters undergo hydrolysis, while only a fraction of aromatic rings are oxidized, and thus the capacity to detect the presence of aromatic systems within the input substrate is not sufficient to successfully predict the occurrence of the corresponding oxidations.

### 2.5. Relevance of the Utilized Descriptors

To explore the relevance of each descriptor included in the utilized sets, the analysis of the feature relevance was performed by Weka. Attention was here focused on the predictive analyses involving the 18 analyzed classes since they represent a comprehensive analysis of all metabolic reactions. The feature relevance was analyzed by considering together all the utilized descriptors. The results were assessed by cross-validation repeated 10 times. Only the descriptors included by at least five validation runs were considered as relevant features.

Appendix A and Figure 2 report the obtained results and evidence that a significant number of descriptors are never relevant in the developed predictive models (31 out of 81). The obtained relevance for the three sets of included descriptors highlights that the physicochemical descriptors show the highest fraction of never-utilized parameters (11 out of 25) followed by the Kier–Hall indices (17 out of 44), while only three stereo-electronic descriptors are never included. The modest role of the physicochemical descriptors can be explained by considering that several features are variously related to molecular size and shape, which conceivably play a limited role in determining the metabolic reactivity. The never-included Kier–Hall indices emphasize that not all the encoded chemical fragments are susceptible to metabolic reactions, while the relevance of the stereo-electronic descriptors confirm the reliability of such features to encode for the chemical reactivity of a given molecule.

Figure 2 reports the occurrence of the frequently relevant descriptors (namely those which are evaluated as relevant in at least 3 predicted classes out of 18). They include 4 physicochemical descriptors, 5 stereo-electronic parameters and 11 Kier–Hall indices. As expected, the relevant physicochemical descriptors include lipophilicity, which is the key factor governing the propensity of a given molecule to be metabolized. The number of H-bond donor groups and the PSA value can encode for both polarity and the presence of chemical groups susceptible to metabolism. Of the five frequently involved stereo-electronic features, three are related to the overall stability/reactivity of a given molecule (dipole, heat of formation and absolute hardness) and two account for its nucleophilicity/electrophilicity profile (E_HOMO and electronegativity). The 11 relevant Kier–Hall indices correspond to the atom types characterizing the reactive groups involved in several metabolic reactions. The most frequently included descriptors comprise the hydroxyl function (sOH), which is engaged by almost all conjugations, and the primary amino group (sNH2), which is involved in both conjugations and redox reactions, while the methyl group (sCH3) and the aromatic carbon atoms (aaCH) are mostly affected by redox reactions. The number of relevant descriptors varies among the predicted classes, ranging from 2 for *O*-glucuronidations to 15 for redox reactions of >NH, >NOH, and –N=O. Although there is no correlation between the number of relevant descriptors and the performance of each class, one may observe that the average number of descriptors for the three main classes is in line with the average performance (redox = 10 descriptors, hydrolyses = 6.3 and conjugations = 7.1; overall average = 8.2), thus confirming that the best-performing models are, on average, characterized by a limited number of variables.

### 2.6. The MetaClass Predictor: An Explorative Study

As detailed in the Methods, the predictive models generated for the classes were deployed by a tool that was developed to predict the occurrence of the corresponding biotransformations for a given input molecule. The tool, called MetaClass, was generated by the Tree2C approach [29] and implemented as an optimized library within the VEGA environment. In this way, this general predictor could directly exploit the VEGA features to calculate the descriptors required by the predictive models. In detail, MetaClass analyzes the input molecule loaded in the workspace of the VEGA program and returns the predicted occurrence for each class.

To provide a preliminary validation of its predictive performance, the MetaClass predictor was tested to predict the metabolic reactions for a set of 10 molecules not included in the database used to develop the models since their metabolic studies were published after 2015. Table 5 summarizes the obtained predictions and reveals encouraging results, especially when considering that most predicted reactions (17 out of 23) belong to redox classes, which provided not so satisfactory models (see Table 3). In detail, Table 5 reveals that the vast majority of reported metabolic reactions (18 out of 23) are conveniently predicted and only three molecules evidence two unrecognized reactions. In addition, all molecules show false positive reactions: this is a rather common issue, which is partly related to the uncertain definition of substrates and non-substrates, as discussed above. In fact, and although the selected papers report rather exhaustive metabolic studies, one cannot exclude a priori the possibility that the false positives might correspond to reactions that can occur but are not reported in the study for various reasons. Hence, we believe that the most significant result of this preliminary validation is the capacity of the MetaClass predictor to conveniently identify most of the reported reactions with a reduced number of unpredicted cases, as assessed by a sensitivity = 0.78. Overall, the confusion matrix compiled in Table 5 leads to an MCC value equal to 0.44 and accuracy equal to 0.80. Not surprisingly, the MCC value is superimposable to the MCC average for the redox classes in Table 3, thus emphasizing the limiting role that the prediction of redox reactions also has for the MetaClass predictor.

**Table 5 molecules-26-05857-t005:** Results of the predictive analysis as obtained by applying the MetaClass predictor to an external set of 10 molecules.

Drug	Reported Classes	TP	TN	FP	FN	Ref.
Atomoxetine	01, 02	2	14	2	0	[30]
Axitinib	01, 02, 05, 08, 22	5	11	2	0	[31]
Benzbromarone	02	1	13	4	0	[32]
Bosentan	01	1	15	2	0	[33]
Brivaracetam	01, 12	2	12	6	0	[34]
Cobimetinib	01, 02, 12	1	13	2	2	[35]
Dasabuvir	01	1	13	4	0	[36]
Efavirenz	01, 02, 22	1	11	4	2	[37]
KAF156	01, 02, 25	2	12	4	1	[38]
Midazolam	01, 22	2	13	3	0	[39]
**Total**	**23**	**18**	**127**	**31**	**5**	**---**

## 3. Methods

### 3.1. Preparation of the Collected Substrates

The study involved 2787 substrates that undergo 3788 first-generation metabolic reactions, as classified into 3 main classes, 21 classes and 101 subclasses. The MetaQSAR database contains the 3D structure of the collected substrates in their neutral state [26]. For the reported predictive analyses, the stored 3D structures were optimized by PM7-based semi-empirical calculations, which also allowed the calculation of an extended set of stereo-electronic parameters [40]. The corresponding ionized forms were generated by an already described script of VEGA [41], which ionizes the molecules by selecting the most probable form at a user-defined pH value (here pH = 7.4). The thus obtained ionized forms underwent the PM7-based optimization procedure as above described. In this study, the tautomeric forms were not considered.

### 3.2. Calculation of Molecular Descriptors

The study involved the calculation of three sets of descriptors. The first set comprised 168 various variables, including constitutional, topological and physicochemical descriptors as computed by BlueDesc. These descriptors were computed by using the Java code available at http://www.ra.cs.uni-tuebingen.de/software/bluedesc/ (accessed on 23 September 2021). In detail, the code was encapsulated into a front-end VEGA script that supports the analysis of molecular databases as well as the generation of the CSV output files.

The second set comprised 25 constitutional, geometrical and physicochemical descriptors as computed by VEGA plus 12 stereo-electronic parameters as calculated by PM7 semi-empirical calculations (see above). The last set comprised 44 Kier–Hall topological indices, which are categorical values that encode for the occurrence of specific functional groups [42]. They were calculated by a specific script, which is included in the standard VEGA release and which detects the occurrence of defined functional groups according to a set of criteria or fingerprints. Specifically, the script analyzes all supported databases and generates a CSV output file.

The complete list of the Kier–Hall topological indices with the corresponding SMARTS strings can be found at https://cdk.github.io/cdk/1.5/docs/api/org/openscience/cdk/qsar/descriptors/molecular/KierHallSmartsDescriptor.html (accessed on 23 September 2021). A description of the included constitutional, geometrical, physicochemical and stereo-electronic descriptors is provided in Appendix A.

### 3.3. Model Building

As mentioned above, the presented predictive studies involved the three major classes and the classes and subclasses including at least 50 metabolic reactions. The initial predictive studies on the three major classes were exploited to select the optimal classification algorithm and to optimize the corresponding parameters. Appendix A lists the MCC values reached for the three major classes by a variety of predictive algorithms implemented in Weka [43]. Although the employment of Tree2C for the development of the MetaClass predictor limited us to the use of tree-based approaches, this analysis also comprised other different methods for the sake of completeness. Appendix A shows that the tree-based algorithms afford, on average, the best performance; in particular, the random forest (RF) classification method [44] provides the highest MCC average value. In detail, RF yields the best MCC value for redox and hydrolytic reactions, while three tree-based algorithms (forestPA, LMT and SysFor) produce slightly better MCC values for conjugations. On these bases, RF was used for all classification analyses reported in this study.

The tuning of the RF hyperparameters was performed by means of the “Experimenter” module in Weka. In detail, the following parameters were considered: (1) the batch size; (2) the number of threads; (3) the number of iterations; (4) the attribute importance. Overall, 14 runs were carried out based on 10-fold cross validation, by which each test was repeated on the dataset 10 times with different random number seeds.

Based on these tests, all the reported classification models were developed by the random forest machine learning algorithm as implemented in the Weka program with the following parameters: (1) the batch size = 100; (2) the number of threads = 1; (3) number of iterations = 100; (4) the attribute importance was not evaluated.

The most significant features were selected by using the Weka program according to both the BestFirst search algorithm (direction = Forward; lookupCacheSize = 1; searchTermination = 5) and the WrapperSubsetEval attribute evaluator (classifier = RandomForest with default settings; doNotCheckCapabilities = False; evaluationMeasure = accuracy, RMSE; folds = 5; seed = 1; threshold = 0.01). As described in the Results, the predictive models were developed by considering the substrates in their neutral form for the three main classes of metabolic reactions and for all the classes and the subclasses with at least 50 occurrences. All the reported predictive models were developed by using balanced datasets as obtained by random undersampling using an ad hoc script implemented in the VEGA program. In contrast, the protonated forms of the substrates and the unbalanced datasets were utilized only to predict the occurrence of the three main classes.

### 3.4. MetaClass Predictor

The models generated for the classes of metabolic reactions (see Table 2) were utilized to develop a predictive system called MetaClass, based on the Tree2C approach [29]. Briefly, Tree2C deploys a tree-based model as generated by the Weka program by converting its output file into a source code by implementing a user-defined programming language. The generated source code also comprises the code required to calculate the included descriptors by exploiting the features already implemented by the VEGA suite of programs. In this way, Tree2C can be utilized to develop scripts, which run within the VEGA environment and which calculate on-the-fly the descriptors required to apply the predictive model to the molecules loaded in the VEGA workspace.

To develop the MetaClass predictor, the Tree2C approach was applied to the predictive models for the metabolic classes by generating the corresponding C code for each model. The generated codes were then merged into a unique script that predicts the occurrence of all the classes for the molecule loaded in the VEGA workspace. To speed up the MetaClass predictor and to support its release, the resulting codes were assembled into a precompiled binary library, which was optimized for use within the VEGA environment.

To simplify the generation of the MetaClass predictor and to support its constant updating, a script to automatize the model generation (MetaClass builder) was also developed. Based on a database of metabolic reactions structured as implemented by MetaQSAR, the script performs the following tasks: (1) extracting the datasets for the classes of metabolic reactions; (2) balancing them by random undersampling of the majority class; (3) calculating the selected descriptors for the collected substrates, which can be calculated directly by the VEGA program or can be provided externally; (4) generating the predictive models by interfacing the script with the Weka program; (5) transforming the models into the corresponding C codes by using Tree2C and (6) assembling the final precompiled library.

MetaClass builder can have two major applications. First, we use this to automatically generate updated releases of the MetaClass predictor, which parallels our continuous efforts to update the MetaQSAR database. Second, any research group can use this tool to develop its own predictor system by exploiting their internal metabolic data, which should be structured as requested by the MetaQSAR database features.

## 4. Conclusions

The study describes a set of tree-based models to predict the occurrence of specific metabolic reactions. They were developed by using a standard random forest procedure and exploiting rather limited sets of representative descriptors. It should be noted that better models might be generated by using extended sets of descriptors through an optimization procedure specifically tailored to each analyzed dataset of metabolic reactions. Nevertheless, the primary objective of the study was focused on an extensive comparison of the achieved performance when predicting the occurrence of the various metabolic reactions as classified by the MetaQSAR database. Thus, the applied procedures were similarly performed for all datasets, and the possible resulting loss of performance similarly affected all the developed models, thus enabling a reasonably unbiased comparison of all the achieved performance results.

When considering the performance achieved by each predicted class, the obtained results can be summarized as follows:
Even when considering the corresponding subclasses, the redox reactions revealed the poorest performance. These negative results underline the complexity of the factors governing these metabolic reactions and suggest that predicting these reactions requires tailored approaches involving optimized algorithms and/or specifically designed descriptors.The hydrolyses provided almost always the best models and benefited from a detailed clustering of their reactions.When subdivided into classes and subclasses, the conjugations yielded, on average, satisfactory predictive models; this suggests that these reactions can be conveniently predicted by recognizing the involved reactive groups.

While considering the comparative aim of the study, the developed models for the considered metabolic classes provided satisfactory performance (i.e., MCC > 0.5 and AUC > 0.8) in 10 out of 18 classes. Therefore, a global classification system, called MetaClass, was developed within the VEGA environment to predict the occurrence of the various metabolic reactions. Clearly, these predictions should be cautiously taken by considering the here described performance. Nevertheless, they can provide a reasonable picture of the overall stability of the tested molecules as well as of the most probable reactions that they can undergo. Finally, a script (MetaClass builder) was developed to automatize the development of the predictive models and of the resulting classification system. Besides allowing the constant updating of the MetaClass predictor, this can be used by any researcher to develop similar predictive tools by exploiting its own metabolic data.

## Figures and Tables

**Figure 1 molecules-26-05857-f001:**
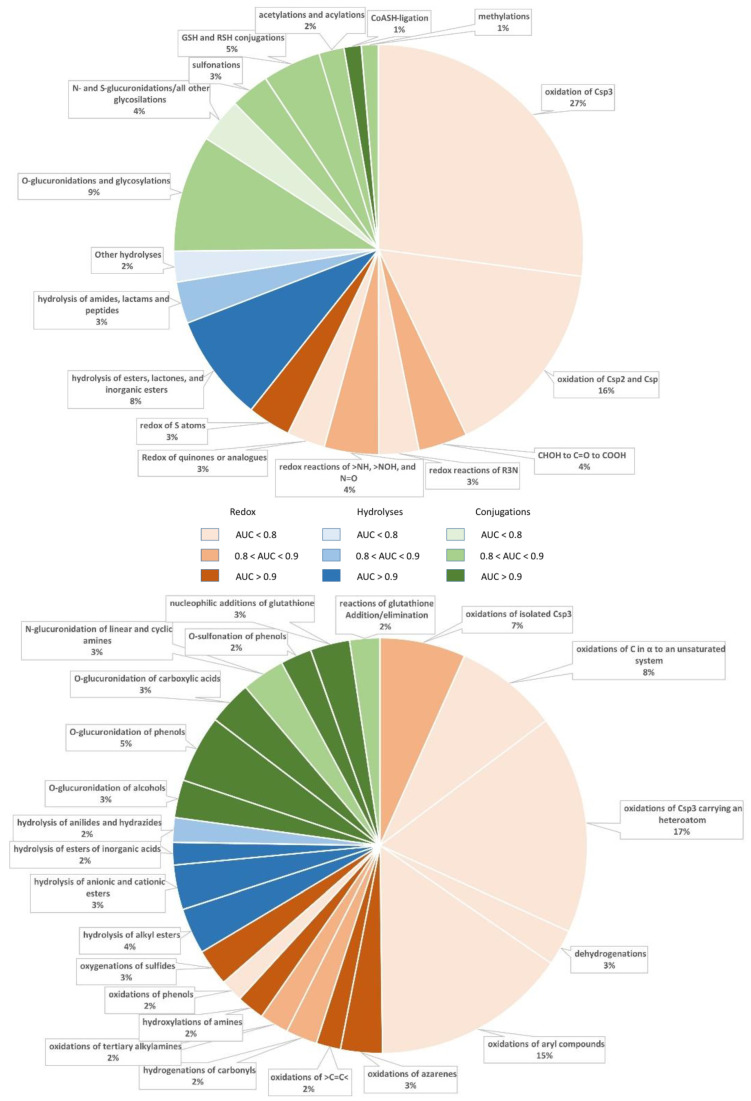
Pie charts representing the predictive performance and relative abundance of the classes (**top**) and subclasses (**bottom**) as subdivided into the three major classes by color coding, detailed in the legend in the middle.

**Figure 2 molecules-26-05857-f002:**
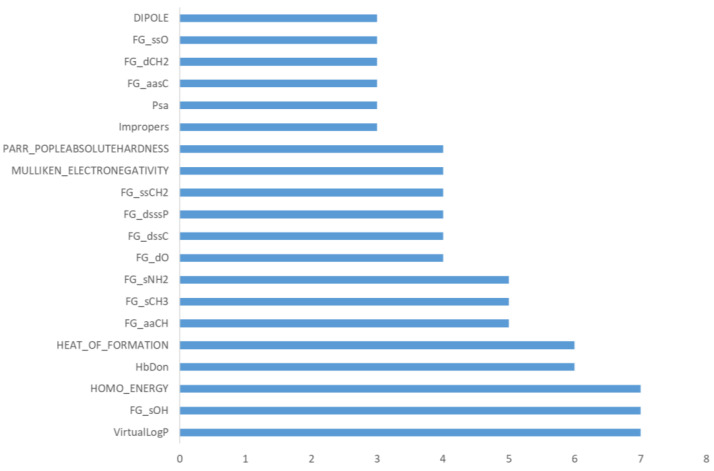
Occurrence of the most frequently relevant descriptors (i.e., those which are relevant for at least 3 classes).

## Data Availability

The MetaClass predictor and the MetaClass builder are available at https://doi.org/10.5281/zenodo.5128531 (accessed on 28 July 2021) as well as all at www.vegazz.net (accessed on 28 July 2021), along with the described scripts. The datasets used in the study are available upon request to the authors. However, the dataset for class 21 is here released as Appendix A as an example of the analyzed input data.

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
