# Peer review of "MetaClass, a Comprehensive Classification System for Predicting the Occurrence of Metabolic Reactions Based on the MetaQSAR Database"

_molecules, 2021, doi:10.3390/molecules26195857_

Round 1

Reviewer 1 Report

The manuscript presents a series of predicted models developed using a relatively sizeable hand-curated database to predict metabolic reactions.

The introduction provides sufficient details. The research design is appropriate. However, methods are not detailed enough, and don t describe all the processes in building the models. Results are extensive but are not clearly presented. The conclusion is too long, and some parts can be included in the results sections.

The following issues need to be clarified:

  1. The authors need to present clearly how the models were constructed.
  2. The internal and external validation methods need to be presented
  3. all data set must be described and attached  if desire to supplemental files so that the whole process can be reproduce by others 
  4. Also, it is customary to present some correlation plots between observed and predictive values

Author Response

Dear Reviewer,

We thank you for your valuable suggestions. Here is a description of the amendments made in the revised version according to your requests.

The introduction provides sufficient details. The research design is appropriate. However, methods are not detailed enough, and don’t describe all the processes in building the models. Results are extensive but are not clearly presented. The conclusion is too long, and some parts can be included in the results sections.

The methods were extended and detailed (see below), results were revised and the conclusions were shortened by deleting considerations already discussed in the Results.

1. The authors need to present clearly how the models were constructed.

The methods utilized to build the models were described in greater detail.

2. The internal and external validation methods need to be presented

The internal and external validation was performed by using the implemented modules in Weka. These processes were carefully reported under Methods.

3. All data set must be described and attached  if desire to supplemental files so that the whole process can be reproduce by others

All data sets are available upon request to the Authors. Due to effort spent to collect these data and considering their intrinsic relevance, we are ready to share them but we would like to retain a certain control of their distribution. 

Also, it is customary to present some correlation plots between observed and predictive values.

Since the presented models are classification models which recognize substrates from non-substrates we cannot draw plots for observed vs. predictive values. We can provides the % of correctly predicted instances and this is encoded by the reported metrics.

We do believe that the revised version fully answers your comments.

Best regards

                                                                                          Alessandro Pedretti

Reviewer 2 Report

The presented manuscript shows new results for predicting relevant metabolic reactions involved in drug metabolism from QSAR descriptions of reaction substrates and drugs utilizing advanced decision tree machine learning methods.  While the methods, results, and interpretation appear to be reasonably thorough, many aspects of the work are not adequately described in the abstract, introduction, and methods, especially within the context of prior literature.

Major issues:

Line 38:  The authors indicate three major applications of AI in drug metabolism, but I would suggest that this statement is short-sighted.  Would suggest the authors change this to “In general, artificial intelligence (AI) methods have at least three potential major applications in drug metabolism…”.  Another major application could be in predicting pharmacokinetics of the drug and its metabolized derivations, just to name another potential major application.

Line 49: Most knowledge-based methods fall under rules-based expert systems, which is a branch of artificial intelligence.  While the statement is accurate, it is misleading, since it implies that knowledge-based methods are not part of classical artificial intelligence field.

Line 76: Something is wrong with the sentence “In these predictive studies, we follow the same classification system adopted by MetaQSAR which subdivides the metabolic reactions into three main classes, classes and 101 subclasses.”  What is the difference between “3 main classes” and “21 classes”.  Maybe you should use a single term for “main class” like “superclass”. Also, the 2018 MetaQSAR paper describes the database as having “101 classes”.  The lack of consistency of terminology is likely to confuse readers.  Also, what is the method of creating this hierarchy of reaction descriptions?  Was it crafted by hand or by some computational method? 

Line 78: What was the specific quantitative criteria for only using the 3 main classes, 18 classes, and 21 subclasses?  This should be added to the introduction or the methods.  Also, the 3 main classes should be specifically listed in the introduction, because it represents the scope of the work.  Also, the abstract should clearly indicate that the “hydrolyses”, “conjugates”, and “redox reactions” represent the “3 main classes”. Likewise, the abstract should mention that the utilized reaction descriptions represented 3788 reactions.

Introduction: The presented research should be put into context of prior research that predicts possible reactions and reaction products.  For instance:

Jeffryes JG, Colastani RL, Elbadawi-Sidhu M, Kind T, Niehaus TD, Broadbelt LJ, Hanson AD, Fiehn O, Tyo KE, Henry CS. MINEs: open access databases of computationally predicted enzyme promiscuity products for untargeted metabolomics. Journal of cheminformatics. 2015 Dec;7(1):1-8.

Hadadi N, Hafner J, Shajkofci A, Zisaki A, Hatzimanikatis V. ATLAS of biochemistry: a repository of all possible biochemical reactions for synthetic biology and metabolic engineering studies. ACS synthetic biology. 2016 Oct 21;5(10):1155-66.

Ryu, Jae Yong, Hyun Uk Kim, and Sang Yup Lee. "Deep learning enables high-quality and high-throughput prediction of enzyme commission numbers." Proceedings of the National Academy of Sciences 116.28 (2019): 13996-14001.

Introduction:  Should include a description of basic description of the three types of descriptors used in the feature vectors with references.  Should add the type of descriptors like “QSAR descriptors”  into the abstract, since “sets of representative descriptors” is completely vague.

Methods:  The authors need to include a better description of the feature vectors generated and used from the three sets of descriptors.  How many features were used in each set?  Which features were encoded numerically versus which ones were encoded categorically?  The authors should really revamp their supplementary Table S2 and include the specific encoding for each feature and which set they belong to.  Would also recommend that the actual training sets be included as supplemental material or as a separate zenodo repository.  If this is already in the zenodo.5128531 repository, then this should be made clear in the manuscript.

Author Response

Dear Reviewer,

We thank you for your valuable suggestions. Here is a description of the amendments made in the revised version according to your requests.

Line 38:  The authors indicate three major applications of AI in drug metabolism, but I would suggest that this statement is short-sighted.  Would suggest the authors change this to “In general, artificial intelligence (AI) methods have at least three potential major applications in drug metabolism…”.  Another major application could be in predicting pharmacokinetics of the drug and its metabolized derivations, just to name another potential major application.

The sentence was accordingly modified.

Line 49: Most knowledge-based methods fall under rules-based expert systems, which is a branch of artificial intelligence.  While the statement is accurate, it is misleading, since it implies that knowledge-based methods are not part of classical artificial intelligence field.

The sentence was accordingly modified.

Line 76: Something is wrong with the sentence “In these predictive studies, we follow the same classification system adopted by MetaQSAR which subdivides the metabolic reactions into three main classes, classes and 101 subclasses.”  What is the difference between “3 main classes” and “21 classes”.  Maybe you should use a single term for “main class” like “superclass”. Also, the 2018 MetaQSAR paper describes the database as having “101 classes”.  The lack of consistency of terminology is likely to confuse readers.  Also, what is the method of creating this hierarchy of reaction descriptions?  Was it crafted by hand or by some computational method? 

The rationale of subdivision of the metabolic reactions into major classes, classes and subclasses was better detailed.

Line 78: What was the specific quantitative criteria for only using the 3 main classes, 18 classes, and 21 subclasses?  This should be added to the introduction or the methods.  Also, the 3 main classes should be specifically listed in the introduction, because it represents the scope of the work.  Also, the abstract should clearly indicate that the “hydrolyses”, “conjugates”, and “redox reactions” represent the “3 main classes”. Likewise, the abstract should mention that the utilized reaction descriptions represented 3788 reactions.

The predictive studies involved the classes and the subclasses including more than 50 instances. This criterion was explained in the Introduction and under Methods.

Introduction: The presented research should be put into context of prior research that predicts possible reactions and reaction products.  For instance:

Jeffryes JG, Colastani RL, Elbadawi-Sidhu M, Kind T, Niehaus TD, Broadbelt LJ, Hanson AD, Fiehn O, Tyo KE, Henry CS. MINEs: open access databases of computationally predicted enzyme promiscuity products for untargeted metabolomics. Journal of cheminformatics. 2015 Dec;7(1):1-8.

Hadadi N, Hafner J, Shajkofci A, Zisaki A, Hatzimanikatis V. ATLAS of biochemistry: a repository of all possible biochemical reactions for synthetic biology and metabolic engineering studies. ACS synthetic biology. 2016 Oct 21;5(10):1155-66.

Ryu, Jae Yong, Hyun Uk Kim, and Sang Yup Lee. "Deep learning enables high-quality and high-throughput prediction of enzyme commission numbers." Proceedings of the National Academy of Sciences 116.28 (2019): 13996-14001.

The first two references were added in the Introduction, the third which refers to a deep-learning method able to predict the EC code of an enzyme starting from its primary sequence (albeit really interesting) appears to be not related to the field of drug metabolism prediction.

Introduction:  Should include a description of basic description of the three types of descriptors used in the feature vectors with references.  Should add the type of descriptors like “QSAR descriptors”  into the abstract, since “sets of representative descriptors” is completely vague.

The sentence in the Abstract was modified accordingly.

Methods:  The authors need to include a better description of the feature vectors generated and used from the three sets of descriptors.  How many features were used in each set?  Which features were encoded numerically versus which ones were encoded categorically?  The authors should really revamp their supplementary Table S2 and include the specific encoding for each feature and which set they belong to.  Would also recommend that the actual training sets be included as supplemental material or as a separate zenodo repository.  If this is already in the zenodo.5128531 repository, then this should be made clear in the manuscript.

As requested also by the first reviewer the Methods were better detailed, Table S2 was modified as suggested. All data sets are available upon request to the Authors. Due to effort spent to collect these data and considering their intrinsic relevance, we are ready to share them but we would like to retain a certain control of their distribution.

We do believe that the revised version fully answers your comments.

Best regards

                                                                                          Alessandro Pedretti

Round 2

Reviewer 1 Report

The  revised manuscript can now  be accepted in the present form.

Author Response

Dear Reviewer,

We thank you for your valuable suggestion. Here is the description of the improvements made in this revised version of the article.

Response to Reviewer 2 Comments

The authors have addressed all but one major concern raised by this reviewer.

In the opinion of this reviewer, it is unacceptable to require direct contact for access to the underlying data used to generate the results presented in this manuscript.  The authors would then effectively control who can scientifically reproduce their results.

As explained in the previous report and considering the relevance of the collected data, we prefer to avoid the uncontrolled dissemination. To meet the request of the reviewer, we added the Table S4 in the supporting information, which includes the dataset used to build the model to predict O-glucuronidations (class 21) and allows the reader to reproduce the results obtained for this class. In this way, the reader can reproduce the results obtained by us for class 21 and if that is not enough, we are ready to share all the data on request.

We do believe that the revised version fully answers your comment.

Best regards

                                                                                          Alessandro Pedretti

Reviewer 2 Report

The authors have addressed all but one major concern raised by this reviewer.

In the opinion of this reviewer, it is unacceptable to require direct contact for access to the underlying data used to generate the results presented in this manuscript.  The authors would then effectively control who can scientifically reproduce their results.  

Author Response

(The authors gave the same response as above.)
